# Rockrose Land Management: Contribution of Periodic Harvesting to Increase Value and to Control *Cistus ladanifer* L. Shrublands

**David Franco Frazão** [1,2,3,*], **José Carlos Gonçalves** [4,5,6], **Amélia M. Silva** [3,7] **and Fernanda Delgado** [4,5,6]

1 Alentejo Biotechnology Centre for Agriculture and Agri-Food (CEBAL)/Instituto Politécnico de Beja (IPBeja), 7801-908 Beja, Portugal

2 Mediterranean Institute for Agriculture, Environment and Development (MED) & Global Change and Sustainability Institute (CHANGE), CEBAL, 7801-908 Beja, Portugal

3 Center for Research and Technology of Agro-Environmental and Biological Sciences (CITAB-UTAD), University of Trás-os-Montes e Alto Douro, Quinta de Prados, 5001-801 Vila Real, Portugal

4 Plant Biotechnology Centre of Beira Interior (CBPBI), Quinta da Senhora de Mércules, Apartado 119, 6001-909 Castelo Branco, Portugal

5 School of Agriculture, Polytechnic Institute of Castelo Branco (IPCB-ESA), Quinta da Senhora de Mércules, 6001-909 Castelo Branco, Portugal

6 Research Centre for Natural Resources, Environment and Society (CERNAS-IPCB), Instituto Politécnico de Castelo Branco, 6000-084 Castelo Branco, Portugal

7 Department of Biology and Environment, University of Trás-os-Montes e Alto Douro (UTAD), Quinta de Prados, 5001-801 Vila Real, Portugal

* Correspondence: david.frazao@cebal.pt

**Abstract:** *Cistus ladanifer* L. (Cistaceae) occupies extensive areas as a dominant species (shrublands) or is associated to other major forest typologies in the Iberian Peninsula. *Cistus ladanifer* shrublands are mostly present in oligotrophic lands with little valorisation and management and as they develop over the years (up to 20-years-old) they promote the ignition and perpetuation of fire. To contribute to the proper management and valorisation of such systems, a 5-year-old dense shrubland was evaluated for its labdanum resin, seeds, and biomass productivity using different non-destructive harvest periodicities (annual and biennial) and seasons (early, mid-, and late summer), in a two-year case-study. Annual harvest modality maximized labdanum resin productivity (reaching $230 \pm 50$ kg·ha$^{-1}$·2 years$^{-1}$ at late summer) and photosynthetic biomass productivity. In contrast, a biennial harvest yielded significant amounts of more diversified products. It maximized seeds productivity (reaching $75 \pm 41$ kg·ha$^{-1}$·2 years$^{-1}$ independently of the summer season) and lignified biomass. However, it also reached a labdanum resin productivity of $134 \pm 20$ kg·ha$^{-1}$·2 yearrs$^{-1}$ at late summer and a photosynthetic biomass productivity around two times lower than the annual harvest. In this study, we propose two modalities of periodic harvest to be considered as proper long cycle management practices of rockrose lands. It intends to minimize fire risks, break the vegetation auto-succession mechanism, and increase profit from non-productive lands based on three direct outputs with a myriad of applications and valorisation pathways.

**Keywords:** Cistaceae; labdanum; photosynthetic biomass; productivity; seeds





## 1. Introduction

*Cistus ladanifer* (Cistaceae) is an evergreen semi-deciduous shrub distributed along the western Mediterranean area (south of France, Iberian Peninsula, North Morocco, and Algeria). Subspecies *ladanifer* (*C. ladanifer* subsp. *ladanifer*) shrublands grow in the meridional half of the Iberian Peninsula over siliceous soils and in the occidental half over shales and granitic soils [1].

*Cistus ladanifer* shrubland is one of the most widespread shrub systems in the Iberian Peninsula. It is poor in other plant species and has a tendency to perpetuate instead of

developing into mature vegetation stages [2]. These shrublands are relatively short-lived and when they are 18/20 years old, they are composed mainly of highly lignified and dried biomass shrub land. Therefore, natural conditions are gathered for the ignition and spread of fire [3]. There is a general idea that the exuded labdanum resin, covering principally the photosynthetic biomass, is flammable and thus helps the spreading of fire, although no specific references can be found to support it. Labdanum gum (resin) is registered in the European Chemicals Agency (EC number 946-963-9) as a non-flammable solid, a statement supported by approved testing methods.

Rather than calling it a pyrophyte species, Trabaud [4] classified this species as opportunistic, occupying open disturbed areas without intense vegetative cover and strong competitors. *Cistus ladanifer* communities are known to comprise an early secondary succession stage of areas after perturbations such as typical Mediterranean fires or human activity (e.g., cleaning of areas through cutting or ploughing) followed by abandonment [5–8]. Reproductive success, low nutritive demands, and resilience/tolerance to stress factors [9] explain the regeneration and perpetuation of such systems.

According to [10], these shrubs improve soil quality in the 0–5 cm top-soil layer, through litter-fall, by enhancing the organic matter and nutrient content beneath their canopies. According to the authors, soil recuperation after a perturbation may favour the posterior establishment of more demanding woodland species such as *Quercus* species, following a typical secondary succession mechanism. However, the resilience of these populations is well demonstrated by their auto-succession process potentiated by the persistent soil seed bank, allelopathy, high plant cover, and relatively short period perturbations [9].

Besides from preventing a more advanced land degradation and protecting wildlife, *C. ladanifer* shrublands have recently been regarded as important carbon sinks, which is important because a key mitigation strategy for climate change is the sequestration of gases such as $CO_2$ [11–13]. In addition, it has been discussed that *C. ladanifer* may have a relevant role to occupy trace-element-contaminated soils by producing heavy metal free biomass or for the phytoextraction of Zn and Mn [14].

The agro–forestry–pasture system known as "montado" in Portugal or "dehesa" in Spain is a "multi-layered dryland system" where evergreen oaks associate with shrubs and native pastures or crops, and from which several benefits and services are supposed to be obtained in a sustainable manner, such as animal raising, hunting, and firewood production [11]. However, shrubs such as *C. ladanifer* must be controlled to not dominate the system. In a 20-year study, Mendes et al. [2] observed that continuous cutting and grazing were able to modify the *C. ladanifer* shrubland in order to progressively and simultaneously reverse land degradation and biodiversity improvement, pointing out grazing as the more economically viable solution because of the cutting costs with no profit.

Based on Godinho-Ferreira et al.'s [15] and Montero et al.'s [16] reports about Portugal and Spain, respectively, three general types of *C ladanifer* shrublands may be proposed:

- Dense shrublands (248,382 and 460,088 ha in Portugal and Spain, respectively): most likely abandoned areas with no management and profit with a high probability of shrubland perpetuation due to cyclic fire events. (Cover index of 0.85 in Godinho-Ferreira et al. [15] and of 0.68 in Montero et al. [16] both considering the whole vegetation but with a dominance by *C. ladanifer*).
- Shrublands associated to oak, pine, or eucalyptus forests (1,783,666 and 2,450,857 ha in Portugal and Spain, respectively): most likely managed areas because of the expected profit (e.g., periodic total clearings of the shrub layer). Open oak woodlands would include the "montado" agro–forestry–pasture system.
- Shrublands associated to a diverse forest (1,111,669 ha in Portugal): most likely poorly managed and with no profit and some probability of dense *C. ladanifer* shrublands reclamation after a fire event.

Hernández-Rodríguez et al. [17] proposed long cycle total clearings (24 years) as a suitable management practice for *C. ladanifer* shrublands based on incomes from biomass harvesting and mushroom (*Boletus edulis*) picking. In fact, *C. ladanifer* biomass, mainly the wood

fraction, may be considered for pellets [18], biochar and biogas [19], and bioethanol [20] production. Biomass may also be transformed into lignin-derivatives and glucan rich solids to be used in bioconversion processes [21], such as lactic-acid production [22] and valorised as a source of cellulose with wood pulp such as crystallinity [23]. Leaves and soft stems as well as condensed tannin extracts have been pointed out for ruminant feeding to some extent [24–26]. However, such a management practice constitutes a high fire risk as pointed out by Oria-de-Rueda et al. [3], because of the years of growth. In addition, indirect economic activities, such as hunting and beekeeping, and direct products such as labdanum resin, or essential oil, capsules, and seeds, have not been considered.

The nutritional value of *C. ladanifer* seeds was addressed in a recent study [27]. Capsules material may be valorised as biofuel [28] or considered for ruminant and monogastric feed in "montados" extensive animal production systems [29].

Labdanum is the resin exuded and covering photosynthetic biomass of *C. ladanifer* [30]. It is nowadays mostly valorised in the perfumery and fragrance industry but has the potential to be used in the cosmetic and pharmaceutical industries given its known bioactivities [14,31]. During summer, one of the traditional and most used harvesting practices involves the harvest of resin productive biomass from which the resin is then extracted using alkaline warm water and acidic precipitation/flotation, a process known as the Andalusian process [14]. This harvest practice is non-destructive of the shrubland and may reconcile with the management practice proposed by Hernández-Rodríguez et al. [17].

The management of *C. ladanifer* may address a number of UN-defined sustainable development goals, including achieving zero hunger (through the production of seeds and roughage), good health and well-being (through the production of labdanum resin), accessible, affordable, and clean energy (through the production of biomass), decent employment and economic growth (through the development of new value and production chains), responsible consumption and production (through the management and effective use of this natural resource), and climate action (by reducing $CO_2$ emissions from wildfires and the loss of carbon sinks) and life on land (restore degraded forests, stop desertification, and maintain/increase biodiversity).

Focusing on dense shrublands in abandoned areas, the aim of this study was to evaluate the effect of the traditional harvesting practice for resin collection while simulating a machinery operation with a constant cut height in a natural *C. ladanifer* shrubland. This case study should contribute to find a suitable management practice for rockrose lands based on the products labdanum resin and seeds productivity, besides biomass. A management technique that does not encourage desertification is thought to be justified by including added-value items as outputs, adding value to unproductive regions.

## 2. Materials and Methods

### 2.1. Characterisation of the Study Area

The natural *C. ladanifer* shrubland selected to carry out the study is situated in Penha Garcia, Idanha-a-Nova, Castelo Branco, Portugal (0.83 ha; GPS coordinates in DMS: 40°01′01″ N 6°59′06″ W), land property of *Sociedade Agrícola de Couto de Penha Garcia, Lda*. According to the agricultural operations records, the field selected was carved clean in 2014 during spring, meaning that at the beginning of the study (2019) it was a 4.5/5-year-old shrubland, since germination occurs in the autumn months. According to the soil chart of the region [32], the selected field is in a region classified with Dystri-Epileptic Regosols. The soil is loamy, acidic, with a low level of organic matter (OM) and available phosphorous, medium level of available potassium, and low levels of exchangeable bases (Table 1). The climatic conditions of the region (climatological normal 1986–2015) are characteristic of the Mediterranean climate: annual mean temperature of 15.0 °C with a mean maximum temperature of 21.5 °C and mean minimum temperature of 9.4 °C, total annual mean rainfall of 735 mm concentrated in the autumn, winter, and spring months, with a dry season during the summer months (June–August) when the monthly mean rainfall is lowest and monthly mean temperature is highest [33]. According to the biogeographic typology of Portugal mainland [34], the

study area presents a *Genisto hirsutae-Cistetum ladaniferi* climatophilous vegetation (Western Mediterranean region, Ibero-atlantic super-province, Luso-Extremadurence province, Cacerence super-district), dominated by *C. ladanifer* and *Genista hirsute* Vahl, but the Labiatae species (e.g., *Lavandula stoechas* L.) are also representative.

**Table 1.** Main physiscal and chemical properties of the study's natural *C. ladanifer* shrubland soil.

| Parameters | | Values |
|---|---|---|
| Texture class (%) | Sand | 56.4 |
| | Silt | 24.7 |
| | Clay | 18.9 |
| | OM | 2.36 |
| Available phosphorus ($P_2O_5$ mg·kg$^{-1}$) | | 7 |
| Available potassium ($K_2O$ mg·kg$^{-1}$) | | 99 |
| Exchangeable bases (mg·kg$^{-1}$) | Calcium | 280 |
| | Magnesium | 51.4 |
| | Potassium | 57.4 |
| | Sodium | 8.86 |
| pH ($H_2O$) | | 5.4 |

### 2.2. Delimitation and Evaluation of Experimental Plots

The natural shrubland was divided into square plots of $10 \times 10$ m$^2$ each, using a 4 m wide disc harrow coupled to a farm tractor to clean the space between and delimit the plots. A total of 48 plots were delimited (in excess for posterior selection based on similarity to minimize environmental effects). In each plot, five $2 \times 2$ m$^2$ squares (near the 4 vertices and in the middle of plot) were evaluated, considering the plants with the main trunk inside the square. At the height where each plant was wider, its maximum and perpendicular diameters were measured and used, as a mean diameter, to estimate the cover area considering a circle [35]. After that, the cover areas of all plants were added up to a total cover area which was then expressed as a percentage of cover (% cover) in relation to the $2 \times 2$ m$^2$ square area. A mean % cover was calculated for each plot using the values from the five sampling squares. Other data such as plants height and density were also recorded.

### 2.3. Soil Sampling and Analysis

A composite soil sample was collected at 20 cm depth, using a soil sampling probe, in October of 2019, representing a polygon with a total area of 0.83 ha. The soil sample was analysed for its granulometry (Robinson's Pipette Method), pH ($H_2O$) (ISO 10390:2005), organic matter (modified Walkley and Black method), available phosphorous and potassium (Égner method), and exchangeable bases (extraction with an ammonium acetate solution buffered at pH 7.0 and quantified by atomic absorption spectrophotometry) in the soil laboratory of the School of Agriculture of Polytechnic Institute of Castelo Branco, Castelo Branco, Portugal.

### 2.4. Experimental Design

The experiment was designed with two combined harvest factors: periodicity and season of the harvest. The first factor comprised two levels (periodicities): annual and biennial. The second factor comprised three levels (seasons): harvest at early summer (late June/early July, ES), mid-summer (August, MS), and late summer (late September/early October, LS). Four plots were assigned to each factor combination, and, in total, 24 plots were needed ($4 \times 2 \times 3 = 24$ plots). Based on the evaluation of the initial 48 plots (Section 2.1), all plots with a % cover higher than 60% were included in the study and distributed randomly between harvest modalities: plots over 80% cover area were randomly distributed and, afterwards, plots with a 60%–80% cover were again randomly distributed. The charac-

terization (vegetation parameters) of the study plots is shown in Table 2. No significant differences were found between plots assigned to each factor combination (ANOVA or Welch's ANOVA ρ-value > 0.05), regarding vegetation parameters of Table 2.

**Table 2.** Vegetation parameters of the selected *C. ladanifer* shrubland.

| Mean Plant Height (m) | Mean Plant Cover Area (m$^2$) | Plant Density (Plants·m$^{-2}$) | Cover (%) |
|:---:|:---:|:---:|:---:|
| 1.10 ± 0.29 | 0.872 ± 0.730 | 1.62 ± 1.17 | 76.9 ± 28.9 |

Values presented as mean ± standard deviation ($n$ = 120).

*2.5. Harvesting and Sampling*

The effect of the harvest factors was evaluated in relation to labdanum resin, seeds, and biomass productivity. At each plot, the biomass was harvested using a hedge trimmer (STIHL HS 45) to cut the plants at 0.5 m height which were then collected by hand and joined as bales to be weighted in a semi-industrial balance (BW-M, Libra Weighing Machines, Ltd.; Sheffield, United Kingdom). After weighting, three biomass subsamples consisting of 3–5 branches each were divided by type of biomass: wood biomass, photosynthetic biomass, and capsules. Those samples were weighted using an analytical balance (d = 0.001, *Sartorius*, model ENTRIS323I-1S). Afterwards, wood biomass was discarded whereas photosynthetic biomass was stored in the freezer until labdanum extraction. Photosynthetic biomass comprises both soft/herbaceous shoots and leaves and semi-lignified biomass, covered by resin/exudate. The capsules were smashed to release seeds which were separated by sieving with a metal mesh of 850 μm aperture and weighted.

*2.6. Labdanum Resin Extraction*

Labdanum was extracted, as described in Frazão et al. [31], from four samples from different plots collected at each season and year ($n$ = 4). Mean values were used to calculate the productivity of each plot given the productivity values of photosynthetic biomass.

*2.7. Biomass Water Content*

The water content of lignified, photosynthetic, and capsules (with seeds) biomass was evaluated at mid-summer on the last year of study (2021), by drying three samples at 105 °C in a ventilated chamber until constant weight. This parameter may be variable along the summer and across different years. Therefore, the parameter was only used for discussion purposes.

*2.8. Statistical Analysis*

Data sets were evaluated for normality (Shapiro–Wilk's test) and homogeneity of variances (Levene's test). Some data sets were transformed to improve the homogeneity of variances between data sets: lignin biomass productivity data was transformed using the y = ln(x) function and capsules biomass productivity data were transformed using y = ln(x + 1) function. One-way ANOVA, *t*-test, two-way ANOVA, and post-hoc Tukey tests were used to compare the mean values between data sets. Statistical analyses were performed for a confidence level of 95% (α = 0.05) using the IBM SPSS Statistics 27 Software.

**3. Results**

Mean extraction yields, from photosynthetic biomass, and mean two-year productivities of labdanum resin are presented in Table 3. In general, both were higher when the harvest was conducted later in the summer. Additionally, productivity was higher when the harvest was performed annually.

**Table 3.** Labdanum resin yield (% dw/fw photosynthetic biomass) and productivity (kg·ha$^{-1}$·2 years$^{-1}$ or kg·ha$^{-1}$·year$^{-1}$) of *C. ladanifer* experimental plots harvested at a constant 0.5 m height from 2019 to 2021. After an initial harvest in 2019, plots were harvested biennially (after two years, in 2021) or annually (every year, in 2020 and 2021), at early summer (June, ES), mid-summer (August, MS), or late summer (October, LS). Each combination of harvest periodicity and season was performed on four different plots (*n* = 4), for details on experimental design see methods Sections 2.2 and 2.4.

| | | Season | | |
|---|---|---|---|---|
| | | ES | MS | LS |
| | | Resin yield (% dw/fw) | | |
| Year | 2020 | 6.02 ± 0.82 | 7.49 ± 0.71 | 7.35 ± 0.65 |
| | 2021 | 6.48 ± 0.41 | 6.11 ± 0.91 | 7.10 ± 0.39 |
| | | Resin productivity (kg·ha$^{-1}$·2 years$^{-1}$) | | |
| Periodicity | Biennial | 89.9 ± 11.8 | 119 ± 25 | 134 ± 20 |
| | Annual (*) | 169 ± 33 | 181 ± 36 | 230 ± 50 |
| | | Resin productivity (kg·ha$^{-1}$·year$^{-1}$) | | |
| (*)Year | 2020 | 81.4 ± 19.5 | 89.0 ± 14.2 | 97.0 ± 40.7 |
| | 2021 | 87.8 ± 14.0 | 92.3 ± 22.5 | 133 ± 23 |

(*) detailed annual productivity from the annual harvest.

Labdanum resin extraction yield varied between 6.02 and 7.49% dw/fw (dry weight/fresh weight). A two-way ANOVA revealed the significant effect of the season of harvest (ρ-value = 0.032) and of the interaction between the year and season of harvest (ρ-value = 0.043) on extraction yield. Within harvest seasons, the only significant difference was found between extraction yields at late summer and early summer (ρ-value = 0.026), being higher for the former. However, when the years of harvest were analysed separately, significant differences were only found for the year of 2020, in which the mean extraction yield at early summer was significantly lower than at mid-summer and late summer (ρ-value = 0.017 and ρ-value = 0.032, respectively).

Regarding mean two-year labdanum productivity, the two-way ANOVA revealed the significant effect of the main factors, harvest periodicity (ρ-value = 0.000) and season of harvest (ρ-value = 0.004), but not of their interaction. The mean productivity of the biennial harvest varied between 89.9 and 134 kg·ha$^{-1}$·2 years$^{-1}$ and the mean productivity of the annual harvest varied between 169 and 230 kg·ha$^{-1}$·2 years$^{-1}$, increasing along summer. The harvest at late summer resulted in a significantly higher labdanum productivity than the harvest at early and mid-summer (post-hoc Tukey test ρ-value: ES vs. MS = 0.954; ES vs. LS = 0.014; MS vs. LS = 0.005). The mean annual productivities of annually harvested plots (Table 3) were not significantly affected by the main factors: year (2020 and 2021) and season of harvest, neither by their interaction.

The mean yields from capsules and mean two-year productivities of seeds are presented in Table 4. The values are presented only for the biennial harvest modality because annual harvests were negligible. Seeds represented around 28.5 ± 4.8% fw·fw$^{-1}$ of the capsules weight and presented a mean two-year productivity of 75 ± 41 kg·ha$^{-1}$·2 years$^{-1}$ when a biennial harvest was performed after the initial 2019 harvest. The ANOVA revealed no significant difference between seasons of harvest for the two parameters: yield (ρ-value = 0.065) and productivity (ρ-value = 0.183).

**Table 4.** Seed yield (% fw·fw$^{-1}$ capsules biomass) and productivity (kg·ha$^{-1}$·2 years$^{-1}$) from *C. ladanifer* experimental plots harvested at a constant 0.5 m height from 2019 to 2021. After an initial harvest in 2019, plots were harvested biennially (after two years, in 2021) at early summer (June, ES), mid-summer (August, MS), or late summer (October, LS). Each harvest was conducted on four different plots (*n* = 4) in each season.

| | Season | | |
|---|---|---|---|
| | ES | MS | LS |
| | Seeds yield (% fw·fw$^{-1}$) | | |
| Biennial Periodicity | 33.1 ± 3.3 | 25.2 ± 5.1 | 28.2 ± 1.8 |
| | Seeds productivity (kg·ha$^{-1}$·2 years$^{-1}$) | | |
| | 103 ± 52 | 81.1 ± 36.6 | 49.3 ± 9.9 |

Note: Production of capsules (and seeds) by annually harvested plots was negligible.

Mean biomass productivities are presented in Figure 1 on a fresh weight basis and discriminated by the type of biomass (lignified, photosynthetic, and capsules) as well as total biomass. The two-way ANOVA revealed that the season of harvest and the interaction between season and periodicity of harvest had no significant effect on productivity of any type of biomass. The same holds true for total biomass productivity between biennial (2755 ± 431 kg·ha$^{-1}$·2 years$^{-1}$) and annual (3133 ± 712 kg·ha$^{-1}$·2 years$^{-1}$) periodicities of harvest. In contrast, the mean two-year productivities were significantly different between the periodicities of harvest when discriminated by type of biomass. Lignified biomass mean productivity was three-fold higher for the biennial harvest (750 ± 345 kg·ha$^{-1}$·2 years$^{-1}$) than for the annual harvest (216 ± 97 kg·ha$^{-1}$·2 years$^{-1}$), representing 26.7 ± 10.2% (fw·fw$^{-1}$) and 7.17 ± 5.10% (fw·fw$^{-1}$) of the total biomass productivity, respectively (two-way ANOVA $\rho$-value for the main factor periodicity = 0.000; productivity values transformed as ln(x) to achieve the assumption of equal variances). Photosynthetic biomass productivity was lower for the biennial harvest (1747 ± 346 kg·ha$^{-1}$·2 years$^{-1}$) than for the annual harvest (2902 ± 750 kg·ha$^{-1}$·2 years$^{-1}$), representing 64.1 ± 12.2% (fw·fw$^{-1}$) and 92.3 ± 5.4% (fw·fw$^{-1}$) of the total biomass productivity, respectively (two-way ANOVA $\rho$-value for the main factor periodicity = 0.000). Capsules biomass mean productivity was eighteen-fold higher for the biennial harvest (259 ± 115 kg·ha$^{-1}$·2 years$^{-1}$) than for the annual harvest (14.3 ± 12.5 kg·ha$^{-1}$·2 years$^{-1}$), representing 9.22 ± 3.16% (fw·fw$^{-1}$) and 0.46 ± 0.63% (fw·fw$^{-1}$) of the total biomass productivity, respectively (two-way ANOVA $\rho$-value for the main factor periodicity = 0.000; productivity values transformed as ln(x + 1) to achieve the assumption of equal variances and include null values).

Within an annual periodicity of harvest, annual biomass productivities in 2020 and 2021 were not significantly different regardless of the type of biomass. Mean annual total biomass productivity was 1567 ± 438 kg·ha$^{-1}$·year$^{-1}$, mean annual lignified biomass productivity was 108 ± 86 kg·ha$^{-1}$·year$^{-1}$, mean annual photosynthetic biomass productivity was 1451 ± 434 kg·ha$^{-1}$·year$^{-1}$, and mean annual capsules biomass productivity was 7.13 ± 8.20 kg·ha$^{-1}$·year$^{-1}$.

In mid-summer of 2021, water content of each biomass fraction was: 42.1 ± 1.1% for lignified biomass, 58.3 ± 1.8% for photosynthetic biomass, and 10.2 ± 0.7% for capsules biomass.

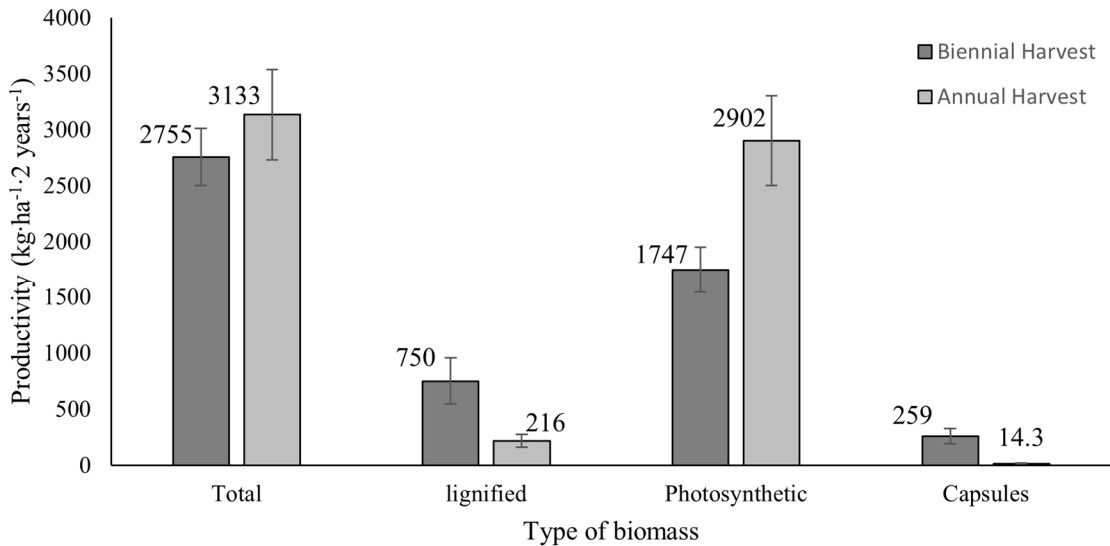

**Figure 1.** Biomass productivity (kg·ha$^{-1}$·2 years$^{-1}$), on fresh weight basis, discriminated by type of biomass, from *C. ladanifer* experimental plots harvested, at a constant 0.5 m height from 2019 to 2021. After an initial harvest in 2019, plots were harvested biennially (after two years, in 2021) or annually (every year, in 2020 and 2021) at early summer (June, ES), mid-summer (August, MS), or late summer (October, LS). Each combination of harvest periodicity and season was conducted on four different plots (*n* = 4). Error bars represent the 95% confidence interval.

## 4. Discussion

On a dry weight basis, labdanum resin (or exudate) extraction yields are reported in literature to be between 5%–15% [30], 7%–18% [36], and 8%–9% [37]. Sosa et al. [30] performed a chloroform extraction. Morgado et al. [36] and Burguer [37] performed the Andalusian process of extraction which is similar to the process used in this study. In the present study, labdanum extraction yields (6%–7%) are reported in relation to biomass fresh weight (Table 3). Considering that the water content of photosynthetic biomass (58.3 ± 1.8%) was constant during the three summer seasons and the two years of study, labdanum extraction yields of this study would rise to 14%–16% on dry weight basis, which match the highest extraction yields reported by the authors reported above.

Labdanum resin productivity during the two years of study was highest when the harvest was conducted later in the summer and annually (Table 3).

The seasonal variation of *C. ladanifer* exudate yield was demonstrated by Sosa et al. [30], who reported a higher yield in the summer and autumn compared to winter and spring. In this study, we further demonstrate that labdanum resin extraction yield varies, overall increasing, through summer (Table 3). However, that variation trend may not be significant for every year, as shown for the year 2020 (Table 3), explaining the significant effect of the interaction between the season and year of harvest on mean resin extraction yield. The quality of the resin was not assessed, although it is reasonable to expect some variation. UV radiation alone or synergistically accompanied with hydric stress was observed to increase the overall content of flavonoid aglycones [38]. Additionally, the same authors further demonstrated that a high temperature and hydric stress increased the content of more methylated flavonoids. According to Alías et al. [39], the content of three labdane-type diterpenoids followed an inverse pattern than that of flavonoids, increasing in winter.

Labdanum resin productivity depends not only on the resin extraction yields, which explains the seasonal variation observed, but also on the productivity of the photosynthetic biomass from which it is extracted, which explains the variation observed according to the periodicity of harvest.

In fact, the mean two-year productivity of photosynthetic biomass was significantly different between harvest periodicities but not between seasons of harvest (Table 3). Two-

year productivity of photosynthetic biomass and of labdanum resin at annual harvesting yielded almost twice the values at biennial harvesting. Morgado et al. [36] report a resin production yield (kg·ha$^{-1}$) of natural shrublands at different ages by the complete harvest of the plants, showing the lowest value of near 230 kg·ha$^{-1}$ from 1.5-years-old shrubland and the highest values of between 400–450 kg·ha$^{-1}$ from 3.5- and 12/15-years-old shrublands. In this study, a single harvest yielded at the maximum 134 ± 20 kg·ha$^{-1}$ (Table 3) and 133 ± 23 kg·ha$^{-1}$ (Table 3) of labdanum resin, in late summer 2021, from plots harvested biennially and annually, respectively. This study aims to evaluate the resin production of existing plants that are non-destructively and continuously harvested at a constant height. At the maximum, given the period of growth before harvest, the resin production yield of annually harvested plots could be compared to 1.5-years-old shrublands and that of biennially harvested plots to 2.5-years-old plots. However, besides the half-year difference of growth, parameters such as vegetation density and cover area may influence resin as well as biomass productivity. Those in turn may vary according to edaphoclimatic, phytosociological, and anthropogenic conditions. In this study, the resin production yield of the 2019 initial harvest was 184 ± 55 kg·ha$^{-1}$ in late summer (data not shown) and at that time the natural shrubland was 4.5/5 years old. A cut at 0.5 m practically harvested all the photosynthetic biomass from the shrubland. Morgado et al. [36] report a resin production yield of around 350 and 370 kg·ha$^{-1}$ from 4.5- and 6/9-years-old shrublands, meaning that the shrubland used in this study was less productive regarding photosynthetic biomass than the shrubland used in Morgado et al.'s study [36].

Contrarily to labdanum resin, the two-year productivity of *C. ladanifer* seeds was higher when the harvest was conducted biennially, however it did not vary with the season of harvest along the summer.

Seeds productivity from plots harvested annually was not registered because capsules biomass productivity was negligible (14.3 ± 12.5 kg·ha$^{-1}$·2 years$^{-1}$), in some plots being null. This is explained by the fact that practically all photosynthetic biomass was harvested upon a 0.5 m height cut, including dolychoblasts that would develop brachyblasts, some of which in turn would develop flower buds and capsules during the following growing season [40,41]. Interestingly, the value for capsules productivity of biennially harvested plots (259 ± 115 kg·ha$^{-1}$·2 years$^{-1}$) is close to the production yield value of the original 4.5-years-old shrubland harvested in 2019 (322 ± 117 kg·ha$^{-1}$, data not shown).

Nuñez et al. [42] reported 2.65 and 1.86% capsules percentage of total biomass and 96 and 299 kg·ha$^{-1}$ of capsules production for 5- and 12/15-years-old shrublands, respectively, on a dry weight basis. Considering the water content of capsules biomass (10.2 ± 0.7%) as a constant for the three summer seasons in this study, on a dry weight basis capsules mean productivity of biennially harvested plots would drop to 233 kg·ha$^{-1}$·2 years$^{-1}$ which is within the range of values reported by Nuñez et al. [42]. The higher percentage in relation to total biomass reported in this study (9.22% fw·fw$^{-1}$) is explained by the fact that Nuñez et al. [42] evaluated the biomass of whole plants whereas in this study only the biomass above 0.5 m height was evaluated. In contrast, the conversion of this percentage, calculated on fresh weight basis, to a percentage on a dry weight basis would rise the value because the water content of photosynthetic biomass (above) and of lignified biomass (41.2 ± 1.1%) are higher than the water content of capsules biomass. At the fresh weight basis, Alves-Ferreira et al. [43] reports that capsules represent between 1 to 4% of the total biomass weight from *C. ladanifer* plants between 5 to 2 years of age. The capsules weight percentage in relation to the total biomass obtained in this study is thus a result of the harvest practice applied. Seeds from the same population under study are reported to have a water content of around 5.88 ± 0.53% in another two-year (2019 and 2020) study [27]. Given the lower water content, the percentage in relation to the capsule's biomass, on a dry weight basis, would also be higher.

In sum, after the initial cut of 4.5/5-years-old shrubland, the annual harvest was shown to be ideal to maximize the photosynthetic biomass and labdanum resin productivity whilst a biennial harvest was ideal to maximize capsules/seed production and lignified

biomass productivity. In addition, the biennial harvest showed a significant productivity of photosynthetic biomass and labdanum resin although around two times lower than the annual harvest.

The annual productivity of a photosynthetic biomass from annual harvested plots was similar to the two-year productivity of biennially harvested plots. It seemed that the photosynthetic biomass produced in one year of growth generated wood biomass during the next year of growth when not harvested while maintaining the productive capacity of photosynthetic biomass and labdanum resin. According to Nuñez et al. [42] and Morgado et al. [36], production of the different types of biomasses from whole plants vary with age until a stabilization point, in general increasing the proportion of lignified biomass. In addition, although photosynthetic biomass percentage in relation to total biomass decreases with age, the total aboveground production increases but only until a certain point. In the two-year period of this study, the same held true for the lignified biomass but not for the photosynthetic biomass which was maintained.

The production yield of the lignified biomass at the initial 2019 harvest was $1860 \pm 766$ kg·ha$^{-1}$ ($49.3 \pm 7.6\%$ in relation to total biomass) at a fresh weight basis. This value is much higher than the productivity of the annual and biennial harvest ($216 \pm 97$ and $750 \pm 345$ kg·ha$^{-1}$·2 years$^{-1}$, respectively). This means that both the annual and biennial periodic harvest reduce the accumulation of fire fuel biomass of the natural shrubland above the cut height.

As mentioned in the introduction section, the harvest of *C. ladanifer* may be separated into distinct products suitable for specific purposes. This approach may increase the value of biomass which does not cover the costs of harvest as shown by Hernández-Rodríguez et al. [17] who had to include the mushroom picking activity to render the harvest more economically viable. In this regard, a new economic study could be conducted contemplating the value of each product but also considering the operation costs such as harvest costs (higher for the annual harvest than for the biennial harvest), biomass separation costs (the annual harvest may have more diversified plant species harvested together with *C. ladanifer* but the biennial harvest may have more diversified type of biomass within *C. ladanifer* biomass), labdanum resin extraction costs, and seeds processing costs (e.g., cleaning and milling). It is also difficult to establish a price for the products from *C. ladanifer* because although extensively studied they are poorly marketed.

A management practice through the harvest of natural shrublands is justified to reduce environmental problems such as fire risks and at the same time to generate value or income from such abandoned forestry areas.

An advantage of exploiting natural shrublands is the low cost "installation" of the productive field; however, a disadvantage is the contamination of harvest biomass with other plant species, increasing post-harvest separation costs. This disadvantage may be more significant for periodic harvesting since it is associated to an increase in biodiversity, however higher for annual harvesting [44]. On the other hand, the fact that periodic harvesting increases biodiversity may mean a break of the auto-succession mechanism of *C. ladanifer* shrublands which in turn may help promote the establishment of more demanding tree species. Vallejo and Alloza [45] discuss strategies for post-fire land restoration in the Mediterranean Basin, highlighting the advantage of sclerophyllous resprout species (such as *Quercus* species) compared to obligate seeders, as is the case of *C. ladanifer*, to provide protection to soil shortly after a fire. According to the authors, oak forests that were never cultivated are reference ecosystems for restoration whereas seeders shrubland communities are often associated to short-term fire cycles and ecosystem degradation loops. However, in this study we have shown that the proper management of *C. ladanifer* shrubland reduce the production of wood biomass, fuel biomass for fire ignition, while maintaining soil cover and generating a possible income. In addition, *Quercus* species such as young cork and holm oaks species were present in the shrubland under study, which means that if properly managed that shrubland may succeed into an oak forest.

As a final remark, *C. ladanifer* shrublands across the Iberian Peninsula are very diverse in their characteristics because of several factors such as age, phytosociology, anthropology, edapho-climatic conditions, among others, which most certainly impact their productivity. Studies on productivity prediction are thus essential.

There are some studies on biomass production and productivity prediction of whole *C. ladanifer* plants and shrublands [11,16,35] but none on the production and productivity of the different types of products. Furthermore, although this case-study is an important first step to propose a suitable management practice for dense shrublands of *C. ladanifer*, there is a need to replicate it for a longer time and with different shrublands (e.g., different age). The combination of the exploitation of *C. ladanifer* may even be thought to make sense during the first years of oak, pine, or eucalyptus plantations although legal frameworks, such as the maximum permitted height of the bush layer for financed forestry or agriculture projects/activities, must be considered.

## 5. Conclusions

The present study demonstrated that periodic and non-destructive harvest inspired by the traditional way of harvest is suitable to manage *C. ladanifer* shrublands, reducing fire fuel biomass, maintaining soil cover, and creating profit from unproductive lands. If the goal is to extract resin, the harvest must be performed every year (annually), yielding almost exclusively photosynthetic biomass. If the goal is to obtain seeds and extract resin, the harvest must be performed every two years (biennially), yielding both photosynthetic and lignified biomass. This management practice may be hypothetically used together with a long cycle management of rockrose lands to yield mushrooms, biomass, labdanum resin, and capsules/seeds. However, it is relevant to replicate this case-study on shrublands with different ages and densities, for longer periods of study, and longer periods of harvest periodicities. Following on those studies, an economic approach may reveal the potential of the periodic harvest and justify the development of suitable forestry machines.

**Author Contributions:** D.F.F.: conceptualization, methodology, formal analysis, investigation, resources, data curation, writing—original draft, and visualization. J.C.G. and A.M.S.: resources, writing—review and editing, and supervision. F.D.: conceptualization, methodology, resources, writing—review and editing, and supervision. All authors have read and agreed to the published version of the manuscript.

**Funding:** This work was supported by the Regional Operational Program of Centro 2020, Portugal 2020, and the European Union, through the European Regional Development Fund (FEDER): CULTIVAR project (CENTRO-01-0145-FEDER-000020), by FCT—Portuguese Foundation for Science and Technology, under the Grant PB/DB/135330/2017 (to D.F.) and under the project UIDB/04033/2020 (CITAB) and by CERNAS/IPCB FCT (Project UIDB/00681/2020).

**Data Availability Statement:** Research data may be accessed upon request made to the authors.

**Conflicts of Interest:** The authors declare that they have no known competing financial interest or personal relationship that could have appeared to influence the work reported in this paper.

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
