# Peer review of "Rockrose Land Management: Contribution of Periodic Harvesting to Increase Value and to Control Cistus ladanifer L. Shrublands"

_forests, doi:10.3390/f14030638_

Round 1

Reviewer 1 Report

The authors need to attend the following suggestions in order to support the relationship among objective, methodology, results, discussion, and conclusions:

·       To specify with more details the study problem, research justification and benefits/beneficiaries.

·       To support statistically the delimitation and evaluation of experimental plots (2.2)

·       Why was collected a composite soil sample at 20 cm depth, using a soil sampling probe, in October of 2019, representing a polygon with a total area of 0.83 ha.?. Argue it. (2.3).

·       Why was designed the experiment with two combined harvest factors: periodicity and 180 season of the harvest). (2.4). Support it statistically

·       To support the harvesting and sampling actions (2.5)

·       The discussion requires more strong support with others researchers and experiences related to the topic research in order to strength the discussion and conclusions, including more international and national references.

·       They need to discuss more with criteria of sustainable development.

·       In the conclusions, they need to support why they are concluding that the possible profits for forest producers are also increased.

·       What is the true contribution to scientific knowledge and the solution of the real problem?

·       What are the practical uses of the results obtained from this study?

·       To specify which were the main limitations to carry out this research?

·       It is necessary to include multidimensional strategies proposals to guarantee the aplication of th obtained results in order to contribute to sustainable development.

·       They need to discuss, what it would happen if the problem is not attended, considering social, ecological, economics, legal and environmental policy impacts.

Author Response

Dear Reviewer,

We kindly appreciate the reviewer’s comments and revision. Justifications and changes to the manuscript according to reviewer’s comments are presented below:

  • To specify with more details the study problem, research justification and benefits/beneficiaries.

Authors’ response: According to reviewer suggestion, the following sentence was added at the end of the Introduction section: “The inclusion of added-value products as outputs is believed to justify a management practice which do not promote desertification, creating a sustainable value in non-productive lands.”

  • To support statistically the delimitation and evaluation of experimental plots (2.2)

Authors’ response: The evaluation of the cover area was based on scientific literature: https://doi.org/10.1016/j.jaridenv.2008.12.009, The reference was cited in section 2.2 and added to the reference list.

Initially, we used a sample size N = 5 (4 m2 subplots as the sampling unit) for the determination of vegetation parameters (e.g. percentage of cover) of each plot (100 m2). Sample represents about 20 % of the plot area. Based on the mean percentage of cover, plots were selected for their homogeneity to minimize environmental effects: From 48 plots 24 were selected.  The following sentence was modified to clarify this issue: “A total of 48 plots were delimited (in excess for posterior selection based on similarity to minimize environmental effects).”

The 10 x 10 delimitation of experimental plots was determined based on cost/time of collecting data and on literature. As an example, Pérez-Devesa et al (2008) (https://doi.org/10.1016/j.foreco.2007.09.074) evaluated a shrubland management in 150 m2 plots in triplicate whereas Prieto et al. (2009) (https://doi.org/10.1007/s11258-009-9608-1) evaluated shrubland management in 20 m2 plots with nine replicates. We used 100 m2 plots with four replicates.

  • Why was collected a composite soil sample at 20 cm depth, using a soil sampling probe, in October of 2019, representing a polygon with a total area of 0.83 ha.?. Argue it. (2.3).

Authors’ response: The 0.83 ha polygon was the area under study. A composite soil sample is done when it is intended to represent the whole polygon area. 20 cm depth of sampling is a common practice on guidelines. Besides, it is representative given the regossol and epileptic characteristic of the soil (soil is an ochric horizon with consolidated hardrock from 25-50 cm of the top soil). All this information is in the manuscript text.

  • Why was designed the experiment with two combined harvest factors: periodicity and 180 season of the harvest). (2.4). Support it statistically

Authors’ response: The experiment was designed to evaluate not only the main effects of periodicity (2 levels) and season (3 levels) of harvest but also their interaction. It was a full factorial experiment based on previous knowledge about the usual collection seasons and the possible periodicities to be done. All information is in the text.

  • To support the harvesting and sampling actions (2.5)

Authors’ response: We believe that the non-destructive harvesting, mimetizing the traditional harvest, is explained in the introduction section. As the experimental unit of the factorial experiment the total harvest weight is the correct approach, as it is in line with the definition of experimental unit. As for the biomass fractions and labdanum resin extraction yield, subsamples were used, generating a mean value and a dispersion value (e.g. standard deviation) for each sample. All further calculations (e.g., productivities) were done, propagating the dispersion measure of the means (standard deviation). 

  • The discussion requires more strong support with others researchers and experiences related to the topic research in order to strength the discussion and conclusions, including more international and national references.

Authors’ response: We appreciate the reviewer comment and suggestion to improve discussion. New added text is marked. And, to the best of our knowledge, we believe that all the studies regarding labdanum resin, seeds and biomass types productivity are referenced. However, there are several other studies addressing biomass estimation but although of value unfortunately they do not evaluate the different biomass types and products as it is done in this study and thus we decided not to include them.

  • They need to discuss more with criteria of sustainable development.

Authors’ response: According to reviewer’s suggestion, a sentence was added to the introduction section: “The management of C. ladanifer may address a number of UN-defined sustainable development goals, including achieving zero hunger (through the production of seeds and roughage), good health and well-being (through the production of labdanum resin), accessible, affordable, and clean energy (through the production of biomass), decent employment and economic growth (through the development of new value and production chains), responsible consumption and production (through the management and effective use of this natural resource), and climate action (by reducing CO2 emissions from wildfires and the loss of carbon sinks) and life on land (restore degraded forests, stop desertification, and maintain/increase biodiversity).”

  • In the conclusions, they need to support why they are concluding that the possible profits for forest producers are also increased.

Authors’ response: According to reviewer recommendation, the sentence was modified to: “…, and creating profit from unproductive lands.” In fact, the beneficiaries from the proposed management may be forest producers, agricultural producers (montado system), governances, land owners, etc.

  • What is the true contribution to scientific knowledge and the solution of the real problem?

Authors’ response: Firstly, we showed that periodic harvest reduced the accumulation of wood biomass which is the fire fuel associated to these shrublands. Secondly, we showed two modalities of harvest with different outputs (sources of income). The problem is the lack of management and of suitable management practices and we present a type of management practice with two possible alternatives.

  • What are the practical uses of the results obtained from this study?

Authors’ response: Regarding the reviewer question, we believe the conclusion section is concise in that respect:

Cistus ladanifer shrublands may be harvested non-destructively to deliver added-value products such as labdanum resin and seeds besides biomass while preventing the accumulation of fire fuel biomass.

  • To specify which were the main limitations to carry out this research?

Authors’ response: Regarding to reviewer question, the limitations of this research work were specified and pointed out in the discussion section. We hope that we have understood the reviewer's question and that the added text has been enlightening.

  • It is necessary to include multidimensional strategies proposals to guarantee the aplication of th obtained results in order to contribute to sustainable development.

Authors’ response: We really appreciate the reviewer comment and we strongly agree with him/her. We propose two harvesting modalities, as mentioned at the end of the manuscript (conclusion), that need to be further validated in longer term studies or forecasting including the economic analysis and ecosystem changes.

  • They need to discuss, what it would happen if the problem is not attended, considering social, ecological, economics, legal and environmental policy impacts.

Authors’ response: We also believe that those impacts are important to address but we are afraid they fall out of the topic of this specific research work. Nevertheless, it is highly relevant to discuss those issues.

Reviewer 2 Report

Dear Authors,

Congratulations. It is well studied work. The subject of the manuscript entitled “Rockrose land management: contribution of periodic harvesting to
increase value and to control Cistus ladanifer L. shrublands.” fits the profile of “Forests” journal. The study delivers some interesting results and can be a source of valuable information. However, the authors made shortcomings that should be corrected and/or revised before the publication of this work.

Abstract

Kindly check minor spelling mistake. The abstract needs a better composition of words.

Introduction

 Authors should explain why they chose Cistus ladanifer L in this work.

Material and methods

Insert the area studied in this experiment if possible.

Results

Results are well indicated.

Discussion

Could be more detailed. The lines should be more focused for highlighting the important finding of this work.

References

Strengthen the part of the discussion with 2 or 3 new references.

Conclusion

Minor language issues must be addressed to improve quality of the MS.

Author Response

Dear Reviewer,

We appreciate the review and the comments done on our study, the changes made accordingly and the justifications follow below.

Abstract

Kindly check minor spelling mistake. The abstract needs a better composition of words.

Authors’ response: According to reviewer comment, the Abstract was modified/corrected.

Introduction

Authors should explain why they chose Cistus ladanifer L in this work.

Authors’ response: We understand the pertinence of this question. This plant was chosen because it covers part of Iberian-Peninsula forests and it has not yet been given the due commercial value. However, we believe that if these forests are well managed they can provide a source of wealth while contributing to the sustainability of the forest. The Introduction of the manuscript explains why this work was done, and for a better comprehension, we added a few new sentences. Cistus ladanifer (lack of management) is the “problem” and we chose to tackle that problem.

Material and methods

Insert the area studied in this experiment if possible.

Authors’ response: The shrubland was to be carved cleaned in that year but an area of 0.83 ha was left for the purpose of this study. The initial area was much wider. Although the area is referenced in 2.3 (Soil sampling and analysis) we added it to the 2.1 section.

Results

Results are well indicated.

Discussion

Could be more detailed. The lines should be more focused for highlighting the important finding of this work.

Authors’ response: The authors appreciated the reviewer’s suggestion and paragraphs were separated to highlight the important findings of the study.

References

Strengthen the part of the discussion with 2 or 3 new references.

Authors’ response: We appreciated the reviewer suggestion, and the following references were cited and discussed in the discussion section:

“5. Alves-Ferreira J, Duarte LC, Fernandes MC, Pereira H, Carvalheiro F (2019) Hydrothermal treatments of Cistus ladanifer industrial residues obtained from essential oil distilleries. Waste and Biomass Valorization 10 (5):1303-1310. https://doi.org/10.1007/s12649-017-0127-3”

“11. Castro H, Freitas H (2009) Above-ground biomass and productivity in the Montado: From herbaceous to shrub dominated communities. J Arid Environ 73 (4-5):506-511. https://doi.org/10.1016/j.jaridenv.2008.12.009 “

Conclusion

Minor language issues must be addressed to improve quality of the MS.

Authors’ response: The MS was further revised in that respect.

Reviewer 3 Report

Dear Authors, 

I congratulate your research and you have made a nice and clear experiment with a well designed and planned so as to let it provide concise information for the applying units/agencies. My only concern is the representativeness of your experiment plots in terms of Cistus ladanifer distribution areas. I am sure you should have a detailed documentation for the studies and the history of similar lands. If you believe that your sites are well representing the densely covered shrublands, please indicate this somewhere in the introduction section. 

One additional issue is regarding the tables 3 and 4. As you may see in the attached MS in pop-up notes, you have provided table 3 and as an expanded data providing table 4 for annual distribution of labdanum productivities. I kindly suggest you to combine these to tables and give significance notes on the values in the table. 

Author Response

Dear Reviewer,

We appreciate your review and the comments which deserved our best attention.

Indeed, C. ladanifer dense shrublands are classified by Godinho-Ferreira (2005) as with a cover index of 0.85 and a dominance index of the tree layer of -0.81. Although the authors do not present the formulas for the index calculation, we assume that cover index refers to cover area and can be transformed into cover percentage (by multiplying by 100), a normal practice in vegetation studies. We also assume that this cover index is calculated for all plants in the area under study and, therefore the specific cover index of C. ladanifer would actually be lower. We believe that is why the authors present the dominance index, in which we believe they compared the cover areas of the important species present herein. However, we cannot know how the dominance index was calculated. The “esteval” of our study had a mean % of cover of about 77 % (Table 2) which is slightly lower than the cover presented by the authors. But the authors cover area (index or percentage) is for the whole plants present in the shrubland and in this study the cover area is representative just of C. ladanifer plant.

Similarly, Montero et al (2020) considered the percentages of cover of each species and compared them to determine dominance. In a different approach they considered a C. ladanifer shrubland (dominated) when the percentage of cover of the species was higher than the others, even in cases where the whole vegetation cover was low (less 10%) which happened on half of the shrublands considered, by the authors, as “jarales de Cistus ladanifer” (Spanish, original language of the study) or “estevais” (in Portuguese). The authors present a mean cover index of the shrubland (fcc) of 0.68. The value for specific C ladanifer cover would probably drop.

For the purpose of righteousness, the cover indexes from Godinho-Ferreira (2005) and Montero et al (2020) were added to text (in the Introduction section).

According to reviewer suggestion, the two table are now joined in the manuscript.

Regarding the pop-up notes in the pdf copy of the manuscript, we really appreciate the careful reading and the suggestions to improve the clarity of the text.

Line 231, I guess this is dry weight/fresh weight. Can you please indicate openly somewhere here?

Authors’ response: The text was revised and the definition was added: dw/fw (dry weight/fresh weight).

Line 250: Are these values sum of 2 years of harvesting for 2020 and 2021 or the harvesting amounts annually?Checking the table 4, I understand that these are 2 year results but this needs to be clarified. You may combine these to tables and give the each year harvests in additional rows or you can make it in your own way.

Authors response:  the values are the sum of the two years. We joined the two tables as requested above and we believe that this issue was clarified using a footnote in the table.

Round 2

Reviewer 1 Report

The authors have satisfied the suggestions that were sent to them, therefore, the manuscript has been sufficiently improved to warrant publication in Forests.